# Epidemiology of Healthcare-Associated Infections and Adherence to the HAI Prevention Strategies

**DOI:** 10.3390/healthcare11010063

**Published:** 2022-12-26

**Authors:** Saleh A. Alrebish, Hasan S. Yusufoglu, Reem F. Alotibi, Nawal S. Abdulkhalik, Nehad J. Ahmed, Amer H. Khan

**Affiliations:** 1Medical Education Department, College of Medicine, Qassim University, Buraydah 51432, Saudi Arabia; 2College of Dentistry and Pharmacy, Buraydah Private Colleges, Buraydah 51411, Saudi Arabia; 3Infection Control Unit, King Saud Hospital, Unaizah 51921, Saudi Arabia; 4Department of Clinical Pharmacy, Pharmacy College, Prince Sattam Bin Abdulaziz University, Al-Kharj 11942, Saudi Arabia; 5Discipline of Clinical Pharmacy, School of Pharmaceutical Sciences, Universiti Sains Malaysia, George Town 11800, Malaysia

**Keywords:** care bundle, HAIs, healthcare-associated infections, incidence

## Abstract

Healthcare-associated infections are widely considered one of the most common unfavorable outcomes of healthcare delivery. Ventilator-associated pneumonia, central line-associated bloodstream infections, and catheter-associated urinary tract infections are examples of healthcare-associated infections. The current study was a retrospective study conducted at a public hospital in Unaizah, Saudi Arabia, to investigate the frequency of healthcare-associated illnesses and adherence to healthcare-associated infection prevention techniques in the year 2021. Surgical site infections occurred at a rate of 0.1%. The average number of catheter-associated urinary tract infections per 1000 catheter days was 0.76. The average number of central line-associated bloodstream infections per 1000 central line days was 2.6. The rate of ventilator-associated pneumonia was 1.1 per 1000 ventilator days on average. The average number of infections caused by multidrug-resistant organisms per 1000 patient days was 2.8. Compliance rates were 94%, 100%, 99%, and 76% for ventilator-associated pneumonia, central line-associated bloodstream infections, catheter-associated urinary tract infections, and hand hygiene bundles, respectively. It is critical to participate in more educational events and workshops, particularly those that emphasize hand cleanliness and personal safety equipment.

## 1. Introduction

Healthcare-associated infections (HAIs) are among the most common unfavorable outcomes of healthcare delivery. HAIs emerge 48 h or more after hospital admission or within 30 days of treatment [1]. Originally, the term “HAIs” referred to infections connected with admission to an acute care hospital (previously known as “nosocomial infections”). HAIs are infections that occur in several locations where patients receive medical care, such as family medicine clinics, long-term care facilities, ambulatory care centers, and residences [2]. Patients at healthcare facilities, particularly hospitals, are affected by healthcare-associated infections that are not noticeable at the time of admission. They also include diseases that afflict healthcare workers and infections that patients acquire while seeking treatment at a hospital or other healthcare facility but emerge after discharge [3].

According to the Centers for Disease Control and Prevention (CDC), healthcare-associated infections include ventilator-associated pneumonia (VAP), central line-associated bloodstream infections (CLABSI), and catheter-associated urinary tract infections (CAUTI). HAIs also include surgical site infections that occur after surgery. The CDC monitors and prevents these illnesses because they pose a considerable risk to patient safety [4]. According to Al-Tawfiq and Tambyah, the most common HAIs include ventilator-induced pneumonia, urinary tract infections from catheters, bloodstream infections from central lines, and surgical site infections [5]. A ventilator user may develop a lung infection known as ventilator-associated pneumonia [6]. When bacteria or viruses enter the bloodstream through the central line, they cause a dangerous infection known as a central line-associated bloodstream infection [6]. A urinary tract infection is defined as any infection of the urinary tract, which includes the urethra, bladder, ureters, and kidney [6]. An infection at the surgical site appears after surgery in the body portion where the procedure was performed [6]. Infections that are resistant to drugs are also included in HAIs [7].

Previous studies reported that seven patients in high-income economies and ten patients in developing and low-income economies had at least one type of HAI for every 100 admitted to a hospital [2,8,9,10,11,12,13]. Healthcare-associated infections affect 4% of hospitalized patients in the United States at any given time [14]. According to the European Center for Disease Prevention and Control, the frequency of HAI in Africa ranges between 2.5% and 14.8%, which is more than double the prevalence in Europe on average [15,16]. HAI affects around 3.2 million people each year, with 37,000 people dying directly from infections that are caused by drug-resistant microorganisms [17,18]. Several Italian investigations [19,20,21] found an HAI incidence rate of 5–10% and a death rate of up to 20–30%.

Healthcare-associated infections (HAIs) affect millions of people each year and affect up to 80,000 patients in Europe every day. This has a significant financial and societal impact [22]. HAIs have been linked to long-term harm, longer hospitalizations, increased rates of antibiotic resistance, additional financial burdens, and even preventable fatalities [17]. Furthermore, previous studies have shown that healthcare-associated infections (HAIs) are a significant cause of morbidity and mortality [2,9,10,11,12,13,23].

Healthcare-associated infections notably increase morbidity and mortality rates, prolonged hospital stays, and increase therapy costs. According to Barchitta et al., about 677 deaths each year in Sicily were attributable to the selected healthcare-associated infection types among people aged 45 years and older [24,25]. Patients who have healthcare-associated infections experience higher mortality rates and longer hospital stays. The quality of life may be impacted by healthcare-associated infections in a similar way to psychological trauma and long-term impairment [24,25].

Hospital environments may harbor pathogenic or opportunistic microorganisms that can infect susceptible people. This is especially concerning in controlled environments with immunocompromised patients and in operating rooms where a variety of risk factors may occur, such as ineffective ventilation systems and healthcare professionals who do not adhere to infection control measures [26]. Monitoring hospital environments is crucial for preventing the spread of healthcare-related infections.

It is critical to apply a variety of prevention techniques to avoid the occurrence of HAIs. According to Collins, it is critical to practice hand hygiene, environmental cleanliness, leadership, proper use of personal protective equipment, consistent evidence-based practices, an antimicrobial-resistance campaign, respiratory hygiene, and evaluation strategies to prevent the occurrence of HAIs [27]. There is a lack of studying the rate of healthcare-associated infections and the adherence of healthcare professionals to the care bundles in Saudi Arabia. The present study is helpful for estimating the occurrence of HAIs in one of the main cities in Saudi Arabia. As a result, this study aimed to investigate the prevalence of healthcare-associated infections and adherence to HAI preventive practices in a public hospital in Unaizah City.

## 2. Materials and Methods

The current investigation was a retrospective study conducted at a public hospital in Unaizah City, Saudi Arabia, to investigate the prevalence of healthcare-associated infections and adherence to HAI preventive efforts. The study comprised infected HAI patients confirmed by the infectious disease department in 2021. Other infections, as well as those that occurred before or during the study period, were excluded.

The patients who had infections in the hospital in 2021 were included from the present study. The information was obtained from the reports of the department of infectious diseases and the infection control unit. The collected data included CLABSI, CAUTI, VAP, SSIs, infections caused by multi-drug resistant organisms (MDRO) and in compliance with HAI prevention methods. The department of infectious diseases and the infection control unit collected information about the number of patients who had one of the HAIs, the number of surgeries that were conducted, and the number of catheter or ventilator days to calculate the rate of different HAIs.

Healthcare-associated infections are defined by the Centers for Disease Control and Prevention (CDC) as infections that people develop while undergoing medical treatment for other diseases. When germs (often bacteria or viruses) enter the bloodstream through the central line, a severe infection called CLABSI develops. A urinary catheter-associated urinary tract infection (CAUTI) is an infection that affects any region of the urinary system, including the urethra, bladder, ureters, and kidney. A person on a ventilator may acquire VAP, a lung infection. An infection that develops at the site of a surgical procedure is known as a surgical site infection.

We calculated the SSI prevalence by dividing the number of SSIs by the total number of surgical cases and multiplying the result by 100%. The rate of CLABSI was calculated by dividing the total number of infections by the number of central line days × 1000. Furthermore, the rate of VAP was calculated by dividing the number of ventilator-associated pneumonia cases by the number of ventilator days * 1000, and the rate of CAUTI was calculated by dividing the number of catheter-associated urinary tract infections by the number of catheter days * 1000. Furthermore, the rate of infections caused by multidrug-resistant organisms was evaluated by dividing the total number of infections by the number of patients days * 1000. The infection control unit prepared a questionnaire that was filled out by healthcare professionals to determine adherence to the recommended bundles. The percentage of adherence to the bundle recommendations was calculated by dividing the number of healthcare professionals who followed the guideline by the total number of participating healthcare professionals * 100%. SPSS was used to analyze the data descriptively to determine the means and the standard deviations.

## 3. Results

### 3.1. The Rate of Surgical Site Infections

Among the 3944 surgeries, only four patients developed surgical site infections. The rate of SSIs in 2021 was 0.1%. The highest rate was in February and May (0.3%). The rate was 0% in eight months (Table 1). The mean rate of SSIs was 0.0833 and the standard deviation was 0.12673.

### 3.2. The Rate of CAUTI, CLABSI, and VAP in the Hospital

The average rate of CAUTI in 2021 was 0.76 per 1000 catheter days, the average rate of CLABSI was 2.6 per 1000 central line days, and the average rate of VAP in 2021 was 1.1 per 1000 ventilator days (Table 2). The mean rate of CAUTI was 0.7083 per 1000 catheter days and the standard deviation was 1.05007. The mean rate of CLABSI was 2.25 per 1000 central line days and the standard deviation was 2.06376. The mean rate of VAP was 1.0583 and the standard deviation was 1.62730.

### 3.3. The Rate of Infections That Were Caused by MDROs

In 2021, 149 patients developed an infection caused by multidrug-resistant organisms. The average rate of MDROs in 2021 was 2.8 per 1000 patient days (Table 3). The mean rate of MDROs was 2.8192 and the standard deviation was 0.89280.

### 3.4. The Adherence to the Care Bundle

The compliance rates for VAP, CLABSI, CAUTI and SSI bundles in 2021 were 94%, 100%, 99%, and 99%, respectively. The compliance rate with hand hygiene practices was 76% (Table 4).

The rate of VAP was 1.5 per 1000 ventilator days, the rate of CLABSI was 2.5 per 1000 central line days, the rate of CAUTI was 0.5 per 1000 catheter days, and the rate of SSI was 0.2% in 2020. On the other hand, the rate of VAP was 1.1 per 1000 ventilator days, the rate of CLABSI was 2.6 per 1000 central line days, the rate of CAUTI was 0.76 per 1000 catheter days, and the rate of SSI was 0.1% in 2021.

## 4. Discussion

This HAI prevalence study was conducted in Unaizah to investigate the prevalence of healthcare-associated infections and adherence to HAI preventive practices in a public hospital in Unaizah city. This is intended to promote the implementation of more evidence-based strategies and practices toward controlling the rate of HAI in healthcare facilities.

The rate of surgical site infections in the present study was 0.1%. According to Ahmed et al., the rate of surgical site infection in a military hospital in Alkharj was 0.41% in 2019 [7]. At a hospital in Makkah, SSI rates were 1.9%, according to a paper by Haseeb et al. [28]. John et al. reported that general surgery and all departments at Sheikh Khalifa Medical City reported SSI rates of 4.68% and 3.57%, respectively [29]. Furthermore, in the surgical department of a general hospital in Malaysia, Wong and Holloway reported that the incidence of SSI was 11.7% [30]. In contrast to the present study’s findings, Alsareii reported that the total SSI rate was 10.2% in a Saudi tertiary care hospital [31]. Khairy et al. reported that in a university hospital in Riyadh, the rate of surgical site infection was 6.8% [32].

The average rate of CLABSI was 2.6 per 1000 central line days. According to Ahmed et al., only 10% of all device-associated HAIs were bloodstream infections linked to central lines [7]. According to Gaid et al., 14.2% of all device-associated HAI were CLABSI [33]. Additionally, Zhang et al. found that the prevalence rate of central line-associated bloodstream infection was 0.63 per 1000 catheter days [34,35]. According to Jahani Sherafat et al., among the device-related HAIs in six academic teaching hospitals in Iran, there were 5.84 central line-associated bloodstream infections (CLABSIs) per 1000 central line days [36]. Additionally, Khan et al. observed that, among 2157 ICU patients in a 760-bed teaching hospital in Eastern India, the mean monthly rate of CLABSI was 1.4 per 1000 device days [10,11]. Additionally, at a general hospital’s intensive care unit (ICU), Iordanou et al. found that CLABSI was the most prevalent device-associated HAI, with a 15.9 incidence rate per 1000 central venous catheter days [37].

The average rate of CAUTI in 2021 was 0.76 per 1000 catheter days. In 2019, there were 1.00 catheter-associated UTIs for every 1000 catheter days in a military hospital in Alkharj, according to Ahmed et al. [7]. Approximately 28.4% of all device-associated HAIs had CAUTI, according to Gaid et al. [33]. In addition, Zhang et al. observed that catheter-associated urinary tract infections were associated with 2.06 per 1000 catheter days [34,35]. In six academic teaching hospitals in Iran, Jahani-Sherafat reported that there were 8.99 catheter-associated urinary tract infections (CAUTIs) per 1000 urinary catheter days among the device-associated HAIs [36]. In addition, Khan et al. observed that among 2157 ICU patients in a 760-bed teaching hospital in Eastern India, the mean monthly rate of CAUTI was 1.25 per 1000 device days [10,11]. Furthermore, Iordanou et al. found that in an intensive care unit at a public hospital in the Republic of Cyprus, the risk of CAUTI was 2.7 per 1000 urinary catheter days [37].

The average rate of VAP in 2021 was 1.1 per 1000 ventilator days. In 2019, there were 2.11 ventilator-associated pneumonia cases per 1000 ventilator days, according to Ahmed et al. [7]. Similarly, Gaid et al. observed that VAP was the device-associated HAI that was most common (57.4%) [33]. Moreover, for every 1000 catheter days, 7.92 ventilator-associated pneumonia cases were reported by Zhang et al. [34,35]. In six academic teaching hospitals in Iran, Jahani-Sherafat reported that there was 7.88 ventilator-associated pneumonia (VAP) cases per 1000 mechanical ventilator days among the device-associated HAIs [36]. Likewise, at a 760-bed teaching hospital in Eastern India, Khan et al. observed that the mean monthly rate of VAP among 2157 ICU patients was 2 per 1000 device days [10,11]. Similarly, the VAP rate in a general hospital in the Republic of Cyprus was 10.1 per 1000 ventilator days, according to Iordanou et al. [37].

In 2021, the average rate of MDROs was 2.8 per 1000 patient days. According to Ahmed et al., there were 3.95 multi-drug resistant organism (MDRO) infections per 1000 patient days in 2019 [7]. According to Balkhair et al., MDRO patients in an Oman teaching hospital provided a prevalence rate of 10.8 MDRO cases per 1000 admissions [38]. Furthermore, Baig et al. reported that there were 4.8 cases per 1000 patient days of hospital-acquired MDROs in a military hospital in Riyadh [39].

There was a low rate of HAIs in the current study because of the correct use of preventative measures and the continued monitoring of inpatients. Al-Thaqafy et al. stated that following the deployment of the ventilator bundle, ventilator bundle compliance drastically increased from 90% in 2010 to 97% in 2013, and the VAP rate decreased from 3.6 cases per 1000 ventilator days in 2010 to 1.0 case per 1000 ventilator days in 2013 [40]. Bagga et al. found a significant decrease in surgical site infections because of the rigorous adherence to the preventive care bundle [41]. Furthermore, Yaseen et al. reported that the use of the HAIs preventive package decreased the rate of CLABSI from 2.0 cases per 1000 central line days to zero cases per 1000 central line days [42]. Prakash et al. reported that the introduction of a care bundle approach reduces the CAUTI rate significantly [43].

The present study focused on compliance with care bundles to prevent the occurrence of HAIs, but there are also other main measures that should be taken into consideration such as the use of antibiotics and the knowledge of healthcare providers about antibiotic use and antimicrobial resistance. Barchitta et al. reported that the use of antibiotics was so common among patients who had healthcare-associated infections that it was the main contributing factor and that patients who received at least one antibiotic had a roughly 19-fold increased risk of infection compared to those who did not. One of the main causes of HAI, particularly for infections brought on by multidrug-resistant bacteria, is the excessive and inappropriate use of broad-spectrum antibiotics, both in healthcare settings and in the community. This highlights the need for initiatives targeted at improving awareness of appropriate antibiotic prescription and reducing the use of broad-spectrum antimicrobials, as well as programs for infection prevention and control and antibiotic stewardship [44,45].

The first limitation of the study is the fact that it was conducted in only one hospital and the results cannot be generalized to other hospitals. To gain a deeper understanding of the true burden HAIs in Unaizah and other cities in Saudi Arabia, larger HAI studies across broader patient populations are required. The second limitation is that the study was conducted during the COVID-19 pandemic, and this could affect healthcare professionals’ adherence to the bundles. Therefore, hospital infection control strategies should be strengthened to improve adherence to care bundles during the COVID-19 pandemic and after the end of the pandemic. The third limitation was that the study focused on care bundles that should be implemented by healthcare professionals and there is no data about the environmental factors such as the air and the surface quality of the hospital wards. Therefore, further studies about environmental factors such as the presence of mold in air and on surfaces are needed.

## 5. Conclusions

The current study found that healthcare-associated infections were uncommon and that the majority of healthcare workers took preventive measures to avoid the occurrence of healthcare-associated infections. Nonetheless, extra training is required, notably in the areas of hand hygiene and personal safety equipment. The hospital’s infectious diseases and infection control unit should ensure that healthcare providers implement healthcare-associated infection prevention activities appropriately.

## Figures and Tables

**Table 1 healthcare-11-00063-t001:** The rate of surgical site infections.

Month	Number of Surgeries	Number of Surgical Site Infections	Percentage
January	296	0	0.0%
February	326	1	0.3%
March	405	0	0.0%
April	314	0	0.0%
May	257	1	0.3%
June	329	0	0.0%
July	221	0	0.0%
August	299	0	0.0%
September	325	0	0.0%
October	308	0	0.0%
November	411	1	0.2%
December	453	1	0.2%
Total	3944	4	0.1%

**Table 2 healthcare-11-00063-t002:** The rate of CAUTI, CLABSI, and VAP.

Month	The Rate of CAUTI	The Rate of CLABSI	The Rate of VAP
January	1.9 per 1000 catheter days	0.5 per 1000 central line days	0 per 1000 ventilator days
February	0 per 1000 catheter days	5.1 per 1000 central line days	3.4 per 1000 ventilator days
March	0 per 1000 catheter days	4.4 per 1000 central line days	0 per 1000 ventilator days
April	0 per 1000 catheter days	0 per 1000 central line days	0 per 1000 ventilator days
May	0 per 1000 catheter days	3.6 per 1000 central line days	0 per 1000 ventilator days
June	0 per 1000 catheter days	3.6 per 1000 central line days	0 per 1000 ventilator days
July	0 per 1000 catheter days	0 per 1000 central line days	4.3 per 1000 ventilator days
August	0 per 1000 catheter days	0 per 1000 central line days	2.6 per 1000 ventilator days
September	2.3 per 1000 catheter days	2.6 per 1000 central line days	0 per 1000 ventilator days
October	2.1 per 1000 catheter days	2.3 per 1000 central line days	0 per 1000 ventilator days
November	2.2 per 1000 catheter days	4.9 per 1000 central line days	0 per 1000 ventilator days
December	0 per 1000 catheter days	0 per 1000 central line days	2.4 per 1000 ventilator days
Total	0.76 per 1000 catheter days	2.6 per 1000 central line days	1.1 per 1000 ventilator days

**Table 3 healthcare-11-00063-t003:** HAIs caused by multidrug-resistant organisms.

Month	Number of MDRO Cases	Number of Patient Days	MDRO Rate per 1000 Patient Days
January	16	4246	3.7 per 1000 patient days
February	15	3793	3.9 per 1000 patient days
March	15	4223	3.5 per 1000 patient days
April	8	4047	1.97 per 1000 patient days
May	5	4272	1.1 per 1000 patient days
June	9	4829	1.86 per 1000 patient days
July	11	4901	2.2 per 1000 patient days
August	15	5106	2.9 per 1000 patient days
September	13	4217	3.0 per 1000 patient days
October	16	4297	3.7 per 1000 patient days
November	15	4273	3.5 per 1000 patient days
December	11	4290	2.5 per 1000 patient days
Total	149	52,495	2.8 per 1000 patient days

**Table 4 healthcare-11-00063-t004:** Care bundle compliance rates.

HAIs	Compliance Rate
The compliance rate with the VAP bundle	94%
The compliance rate with the CLABSI bundle	100%
The compliance rate with the CAUTI bundle	99%
The compliance rate with the SSI bundle	99%
The compliance rate with hand hygiene practices	76%

## Data Availability

Not applicable.

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
