# Peer review of "Epidemiology of Healthcare-Associated Infections and Adherence to the HAI Prevention Strategies"

_healthcare, 2022, doi:10.3390/healthcare11010063_

Round 1

Reviewer 1 Report

I ve read with a great interest the paper entitled "Epidemiology of Healthcare-Associated Infections and the adherence to the HAIs’ prevention strategies in a Public Hospital in Unaizah City". The paper is coherent to the scope of the journal aiming to  to to determine the incidence of healthcare-associated infections in a hospital of Saudi Arabia and the adherence to the healthcare-associated infections’ pre-vention strategies in 2021 
The topic is interesting even for an international reader, but it seems to be just an epidemiological study that should be better analyzed by the authors.

Author Response

Thank you for your comment. The data we have is extracted from the infection control unit report including only percentages without details. So, the statistical analysis is difficult. Nonetheless, we use SPSS 25 to find the means and the standard deviations of the infection rates, and we add these results to the paper.

Reviewer 2 Report

This is an interesting study aiming at describing the occurrence of HAIs in in a Public Hospital in Unaizah City. Although interesting, some suggestions could improve the quality of this manuscript.

In the introduction section, more emphasis should be given to the impact of HAIs worldwide. Beyond prevalence and incidence, additional measures (such as measures of impact) could be considered. Please consider a recent article about the burden of HAIs in Italy for example (doi: 10.3390/antibiotics10111360)

In the method section, more details should be given about case definition. It is not clear if the authors used the ECDC definition for HAIs, that are all the infections diagnosed after two days from hospital admission.

Moreover, the authors should better describes methods for collecting and analysing data.

Regarding preventive measures, the authors reported compliance using invasive devices. While interesting and crucial for HAI prevention, the authors should at least discuss about other factors that are important (for example antibiotic use and knowledge of healthcare workers on AMR). In line with suggestion, please also consider as an example the following articles: doi: 10.3390/antibiotics10010001; doi: 10.3390/ijerph16132253; 

Finally, I would suggest an extensive revision for english style and language

Author Response

This is an interesting study aiming at describing the occurrence of HAIs in in a Public Hospital in Unaizah City. Although interesting, some suggestions could improve the quality of this manuscript.

In the introduction section, more emphasis should be given to the impact of HAIs worldwide. Beyond prevalence and incidence, additional measures (such as measures of impact) could be considered. Please consider a recent article about the burden of HAIs in Italy for example (doi: 10.3390/antibiotics10111360)

We add another paragraph about the burdens of HAIs from the recommended article.

In the method section, more details should be given about case definition. It is not clear if the authors used the ECDC definition for HAIs, that are all the infections diagnosed after two days from hospital admission.

We add the CDC definitions of HAIs, SSIs, CAUTI, CLABSI, VAP.

Moreover, the authors should better describe methods for collecting and analyzing data.

We add data about the methods of collecting and analyzing data.

Regarding preventive measures, the authors reported compliance using invasive devices. While interesting and crucial for HAI prevention, the authors should at least discuss about other factors that are important (for example antibiotic use and knowledge of healthcare workers on AMR). In line with suggestion, please also consider as an example the following articles: doi: 10.3390/antibiotics10010001; doi: 10.3390/ijerph16132253; 

We add a paragraph about the other factors that are important

Finally, I would suggest an extensive revision for english style and language

We modified the English style and language

Reviewer 3 Report

First of all I would like to thank for the opportunity to review this paper. The importance of healthcare associated infection control is well known and emphasized also by the pandemic. Actually, the environmental factors are among the main methods to their counteracting and the role of environmental monitoring is central. In this context, aim of the paper under review is to investigate the incidence of healthcare-associated infections and the adherence to the healthcare-associated infections’ prevention strategies in 2021 in a public hospital in Unaizah, Saudi Arabia.

The subject under study is certainly important, but it is nevertheless believed that the paper must be improved before publication especially for its local impact. I would like to encourage authors to consider several issues to be improved.

Title: it can be improved, being more attractive for the readers.

Introduction: The authors should improve the introduction, making clearer what is the gap in the literature that is filled with this study. The authors must better frame their study within the vast body of literature that addressed the issue of hospital environmental monitoring in the control of healthcare related infection (refer to articles with DOI: 10.1186/s12879-014-0595-z) also at international level. Then they must report the knowledge already existing and that they will consider. Finally, they must show what they want to do and how they want to do it.

Methods: This section is very poor, it must be improved to report all the methods used to get all the results reported e.g. the Authors report the Care bundle compliance rates, how did they get this information? The section on the infections’ prevention strategies is completely missing or unclearly reported.

Moreover, the survey was conducted in only one hospital in 2021. The period was during the second year of the pandemic, was this factor considered? This can be a bias in the results reported.

The enrolment procedure must be specified. How did the authors choose the setting to perform the study? How the results can be referred to a general hospital population in the Authors’ country? A non-representative sample is by its self a non-sense-study.

Discussion: I also suggest expanding. Emphasize the contribution of the study to the literature. The discussion must be updated in light of the different kind of HCAI, especially those due to molds (see the above mentioned reference). The Authors should add more practical recommendations for the reader, based on their findings. Also, the section of limitations and future search is also very short, the Authors could elaborate on that.

Author Response

Comments and Suggestions for Authors

First of all I would like to thank for the opportunity to review this paper. The importance of healthcare associated infection control is well known and emphasized also by the pandemic. Actually, the environmental factors are among the main methods to their counteracting and the role of environmental monitoring is central. In this context, aim of the paper under review is to investigate the incidence of healthcare-associated infections and the adherence to the healthcare-associated infections’ prevention strategies in 2021 in a public hospital in Unaizah, Saudi Arabia.

The subject under study is certainly important, but it is nevertheless believed that the paper must be improved before publication especially for its local impactI would like to encourage authors to consider several issues to be improved.

Title: it can be improved, being more attractive for the readers.

I modified the title

Introduction: The authors should improve the introduction, making clearer what is the gap in the literature that is filled with this study. The authors must better frame their study within the vast body of literature that addressed the issue of hospital environmental monitoring in the control of healthcare related infection (refer to articles with DOI: 10.1186/s12879-014-0595-z) also at international level. Then they must report the knowledge already existing and that they will consider. Finally, they must show what they want to do and how they want to do it.

We add a paragraph about environmental monitoring and also we add a limitation that the study focused on care bundles that should be implemented by healthcare professionals and there is no data about the environmental factors such as the air and the surface quality of the hospital wards. Furthermore, we add a paragraph about the lack of studying the occurrence of HAIs I n Saudi Arabia.

Methods: This section is very poor, it must be improved to report all the methods used to get all the results reported e.g. the Authors report the Care bundle compliance rates, how did they get this information? The section on the infections’ prevention strategies is completely missing or unclearly reported.

Moreover, the survey was conducted in only one hospital in 2021. The period was during the second year of the pandemic, was this factor considered? This can be a bias in the results reported.

The enrolment procedure must be specified. How did the authors choose the setting to perform the study? How the results can be referred to a general hospital population in the Authors’ country? A non-representative sample is by its self a non-sense-study.

We add data on calculating the care bundle compliance rates.

We add a limitation the study was conducted in only one hospital and can’t be generalized to the other hospitals. We add also a limitation that the study was conducted during the COVID-19 pandemic and this could affect the adherence to the bundles.

Discussion: I also suggest expanding. Emphasize the contribution of the study to the literature. The discussion must be updated in light of the different kind of HCAI, especially those due to molds (see the above mentioned reference). The Authors should add more practical recommendations for the reader, based on their findings. Also, the section of limitations and future search is also very short, the Authors could elaborate on that.

We updated the discussion and elaborate more on the limitations, future researchers, and recommendations.

Round 2

Reviewer 1 Report

I appreciate the efforts of the authors however and I can consider correct WThe data we have is extracted from the infection control unit report including only percentages without details. So, the statistical analysis is difficult". I think they should read and include in the references the following reference 

Infections are a very dangerous affair: Enterobiasis and death DOI: 10.3390/healthcare9121641

Reviewer 2 Report

None

Reviewer 3 Report

the paper was improved and it is now suitable for publication